# TLR4 Expression in Ex-Lichenoid Lesions—Oral Squamous Cell Carcinomas and Its Surrounding Epithelium: The Role of Tumor Inflammatory Microenvironment

**DOI:** 10.3390/biom12030385

**Published:** 2022-02-28

**Authors:** Fernanda Visioli, Julia Silveira Nunes, Maria Carmela Pedicillo, Rosalia Leonardi, Angela Santoro, Gian Franco Zannoni, Gabriella Aquino, Margherita Cerrone, Monica Cantile, Nunzia Simona Losito, Vito Rodolico, Giuseppina Campisi, Giuseppe Colella, Ilenia Sara De Stefano, Maria Antonietta Ramunno, Cristina Pizzulli, Marco Visconti, Lorenzo Lo Muzio, Giuseppe Pannone

**Affiliations:** 1Department of Oral Pathology, Universidade Federal do Rio Grande do Sul, Porto Alegre 90035003, RS, Brazil; ju.silveiranunes@gmail.com; 2Pathological Anatomy Unit, Department of Clinical and Experimental Medicine, University of Foggia, 71122 Foggia, Italy; mariacarmela.pedicillo@unifg.it (M.C.P.); ileniadestefano@hotmail.it (I.S.D.S.); ramunno.m.ant@gmail.com (M.A.R.); cristina.pizzulli@unifg.it (C.P.); marcovis88@hotmail.it (M.V.); giuseppe.pannone@unifg.it (G.P.); 3Department of Medical-Surgical Specialties, University of Catania, 95124 Catania, Italy; rleonard@unict.it; 4Department of Sciences of the Women and Child Health, Fondazione Policlinico Agostino Gemelli, Università Cattolica del Sacro Cuore, 00168 Rome, Italy; angelasantoro1@hotmail.it (A.S.); gianfranco.zannoni@unicatt.it (G.F.Z.); 5Pathology Unit, National Cancer Institute, Fondazione “G. Pascale”, Via Mariano Semmola, 80131 Napoli, Italy; gabryaquino@gmail.com (G.A.); mcerrone1@virgilio.it (M.C.); monicantile@libero.it (M.C.); n.losito@istitutotumori.na.it (N.S.L.); 6Department of Sciences for the Promotion of the Maternal and Childhood Health “G. D’Alessandro”, Section of Anatomic Pathology, University of Palermo, A.O.U. Policlinico “P. Giaccone”, 90133 Palermo, Italy; vitorodolico@gmail.com; 7Department of Surgical, Oncological and Oral Sciences, University of Palermo, Via del Vespro 129, 90133 Palermo, Italy; campisi@odonto.unipa.it; 8Department of Head and Neck Surgery, Second University of Naples, 80131 Naples, Italy; giuseppe.colella@unicampania.it; 9Department of Clinical and Experimental Medicine, Oral Pathology Unit, University of Foggia, 71122 Foggia, Italy; lorenzo.lomuzio@unifg.it

**Keywords:** head and neck cancer, tumor microenvironment, inflammation, toll-like receptors, TLR4

## Abstract

Toll-like receptors (TLRs) regulate innate and adaptive immune responses. Moreover, TLRs can induce a pro-survival and pro-proliferation response in tumor cells. This study aims to investigate the expression of TLR4 in the epithelium surrounding oral squamous cell carcinomas (OSCC) in relation to its inflammatory microenvironment. This study included 150 human samples: 30 normal oral control (NOC), 38 non-lichenoid epithelium surrounding OSCC (NLE-OSCC), 28 lichenoid epithelium surrounding OSCC (LE-OSCC), 30 OSCC ex-non oral lichenoid lesion (OSCC Ex-NOLL), and 24 OSCC ex-oral lichenoid lesion (OSCC Ex-OLL). TLR4 expression was investigated by immunohistochemistry and the percentage of positive cells was quantified. In addition, a semiquantitative analysis of staining intensity was performed. Immunohistochemical analysis revealed that TLR4 is strongly upregulated in LE-OSCC as compared to normal control epithelium and NLE-OSCC. TLR4 expression was associated with the inflammatory environment, since the percentage of positive cells increases from NOC and NLE-OSCC to LE-OSCC, reaching the highest value in OSCC Ex–OLL. TLR4 was detected in the basal third of the epithelium in NLE-OSCC, while in LE-OSCC, TLR4 expression reached the intermediate layer. These results demonstrated that an inflammatory microenvironment can upregulate TLR4, which may boost tumor development.

## 1. Introduction

Oral and pharyngeal cancer represent the sixth most common cancers in the world; an estimated 354,900 new cases and 177,400 deaths from oral cavity cancer (including lip cancer) occurred in 2018 worldwide [1]. Despite advances in diagnosis and treatment, oral squamous cell carcinoma (OSCC) continues to have a poor 5-year survival rate [2]. The important role of the tumor microenvironment (TME) in the progression of cancer has become increasingly evident. OSCC may arise in an immune cell-rich environment, where inflammatory cells within TME may produce both pro-cancerogenic and anti-cancerogenic effects [3].

In the past, the immune response was thought to act only as an anti-tumor mechanism, as an attempt to prevent cancer progression. However, updated research has shown that inflammation can also support tumor growth in many ways: inducing proliferation, cancer cell survival, angiogenesis, and favoring tumor invasion [4,5,6]. Moreover, inflammation in the early stages of carcinogenesis can boost the cancer development [6,7]. This is particularly important regarding potentially malignant disorders associated with chronic inflammation and immune activation, such as oral lichen planus and oral lichenoid lesions [8]. Oral lichenoid lesion (OLL) is a term used to diagnose white and/or red, unilateral or bilateral lesions which cannot be characterized as oral lichen planus. Histologically, OLL presents an intense inflammatory infiltrate underlying epithelial tissue, which can be deeper and extend beyond the epithelial-connective tissue interface [9]. 

The modulation of the inflammatory process by toll-like receptors can be a key factor linking inflammation and tumor development [10]. Toll-like receptors (TLRs) are expressed on cells of the immune system as dendritic cells, as well as on non-immune cells, such as keratinocytes. TLRs act as sensors to recognize pathogens and can regulate innate and adaptive immune responses, recruiting immune cells [11,12]. A total of 10 TLRs have been identified in humans. From these, TLR4 has been associated with pro-tumorigenic outcomes [13,14]. Toll-like receptor 4 (TLR4) primarily recognizes and is activated exogenously by lipopolysaccharide (LPS). However, endogenous ligands from damaged tissues, such as high-mobility group box 1 (HMGB1), heat shock proteins (HSP), reactive oxygen species (ROS), peptides derived from fibrinogen, and monosodium urate crystals, were also identified [10,11].

TLR4 activation requires interaction with the auxiliary protein CD14 and the co-receptors myeloid differentiation protein 2 (MD-2), and the downstream signaling cascade is mediated through two different pathways: myeloid differentiation factor 88 (MyD88)-dependent and MyD88-independent. The MyD88 axis results in the activation of the NF-Kβ signaling and, consequently, in an inflammatory pathway with COX-2 activation, as well as the secretion of different cytokines and growth factors, such as IL-1β, IL-6, IL-8, and TNF- α, which will induce a pro-angiogenic, pro-survival, and pro-proliferation response. The MyD88-independent axis, on the other hand, recruits the TRAM adapter which activates TRIF. TRIF with TBK1 and IKKi proteins phosphorylates IRF3 and activates this transcription factor, resulting in the transcription of type I interferons [10,11,12,13,14].

It has been shown that TLR4 is increased, both at gene and protein levels, in oral lichenoid lesions in comparison to normal tissue [15]. Kotrashetti et al. (2013) observed that expression of TLR4 improved with increasing degrees of oral epithelial dysplasia. TLR4 was also strongly expressed in OSCC, suggesting a role of TLR4 in OSCC development [16]. The invasive growth of oral tongue squamous cell carcinoma has been associated with high levels of TLR2, TLR4, and TLR9 [17]. In addition, mice deficient in TLR4 showed decreased risk of developing gastric cancer [18]. However, while TLR4 shows mostly pro-tumor effects, it can also result in interferon expression and secretion, which can induce an anti-tumor response [19].

The present study aims to analyze the expression of TLR4 in the peritumoral epithelium tissue and in OSCC in relation to the inflammatory microenvironment. In addition, it will be investigated whether the origin of a lichenoid lesion influences the TLR4 pattern in OSCC.

## 2. Materials and Methods

This study was conducted in accordance with good clinical practice guidelines and the Declaration of Helsinki (1975, revised in 2013). The clinical information was retrieved from the patients’ medical records and pathology reports. The patients' initials or other personal identifiers did not appear in any image. Finally, all samples were anonymized before histology and immunohistochemistry. The Ethical Committee of the University of Palermo and the Azienda Ospedaliera Universitaria Policlinico Giaccone–Palermo considered the retrospective nature of the study and approved the submission of this scientific work (Ethical committee: Prot. N° 11/2011). Analyzed data were collected as part of the routine diagnoses. The patients were diagnosed and treated according to national guidelines and agreements.

The present study is multicentric, with the involvement of different cancer centers. Cases were retrieved from the files of the pathology units of the Universities of Foggia (Department of Clinical and Experimental Medicine, Pathological Anatomy Unit, OO.RR. Foggia), Rome (Catholic University, Fondazione Policlinico Universitario A. Gemelli, IRCCS), Palermo (A.O.U. Policlinico “P. Giaccone”), Naples (National Cancer Institute, Institute Fondazione G Pascale and Second University of Naples SUN), and Catania (A.O.U. “Policlinico—Vittorio Emanuele” P.O. G. Rodolico).

The tissues were routinely formalin-fixed and paraffin-embedded (FFPE). All tissue slides were reviewed by a certified pathologist (GP). The study included 150 samples: 30 normal oral control (NOC), 38 non-lichenoid epithelium surrounding OSCC (NLE-OSCC), 28 lichenoid epithelium surrounding OSCC (LE-OSCC), 30 OSCC ex non lichenoid lesion (OSCC Ex-NOLL), and 24 OSCC ex lichenoid lesion (OSCC Ex-OLL).

The control group (normal oral mucosa—NOC) consisted of subjects with no known systemic or oral diseases. Biopsies were obtained from normal buccal mucosa of individuals with no clinical history of autoimmune disease. Tumor samples were assessed regarding the non-tumoral epithelium surrounding the tumor (LE-OSCC, NLE-OSCC) and the center of the tumor (OSCC-Ex-OLL, OSCC-Ex-NOLL,). 

OSCC ex-oral lichenoid lesions (OSCC-Ex-OLL) were samples of tumors derived from previous lesions with a diagnosis of oral lichenoid lesion (OLL), according to criteria established by van der Meiji and van der Waal (2003) [20]. OLL diagnosis was defined when clinical or histopathological criteria were not met for oral lichen planus diagnosis [20]. Reactions to dental materials, termed oral lichenoid reactions (OLR), were excluded. OSCC ex-non-lichenoid lesions (OSCC-Ex-NOLL) were tumor samples not derived from previous oral lichenoid lesions.

The epithelium surrounding OSCC with at least 1 cm distance from the tumor was assessed by two pathologists; this was possible since they were tumors resected with margins. The LE-OSCC lesions were defined as OSCC being surrounded by histologically proven oral lichenoid epithelium with absence or slight cytological atypia limited to architectural findings, with chronic phlogistic infiltration lacking the typical aspects of oral lichen planus (OLP) and of oral lichenoid dysplasia, since recent molecular data revealed that oral lichenoid dysplasia is not a distinct pathological entity, being similar to OLP [21]. Non-lichenoid epithelium surrounding (NLE-OSCC) was selected when the epithelium surrounding squamous cell carcinomas showed no inflammatory infiltrate, in neither the superficial chorion, deep chorion, nor lymphocytic exocytosis in the epithelium.

IHC was performed on 4 μm paraffin sections mounted on poly-L-lysine-coated glass slides, by automated linked streptavidin-biotin horseradish peroxidase (LSAB-HRP) technique, performed by Ventana Benchmark^®^ XT autostainer, using a specific monoclonal antibody against TLR-4 (NOVUS BIOLOGICALS, clone 76B357.1, code NB100-56566, dilution 1:300). Representative samples were additionally stained with antibody against phospho-NFKB Ser933 (clone 178F3, 1:100, CELL SIGNALING TECHNOLOGY), COX2 (clone SP21, prediluted, CELL MARQUE, ROCHE), and IL6Rα (clone H-7, 1:50, SANTA CRUZ BIOTECHNOLOGY). Gill’s type II hematoxylin was used for nuclear counterstaining. Appropriate positive and negative controls were run for the tested antibody.

Immunostained slides were acquired by digital camera and analyzed by ISE TMA Software (Integrated System Engineering, Milan, Italy) and CellSens V1.9^®^ Olympus image analysis software. Two of the authors (GP, AS) evaluated the results of the IHC staining separately; in a second step, inter-observer agreement was obtained in case of discordant evaluations.

Cases were evaluated on the basis of percentage of positive cells (0–100). The staining intensity was also evaluated and scored as follows: negative (0), faint (1), moderate (2), strong (3). Finally, intensity staining was multiplied to IHC percentage in order to obtain the final expression score, ranging from 0 to 300 units. TLR-4 localization in oral lesions was scored as basal (1), basal + intermediate (2), and full-thickness epithelium (3). 

All data were analyzed by MedCalc 13.0.6.0 (for Windows) and SOFA Statistics 1.4.5 (for Linux) statistical software Debian 8.2 and Windows Operating Systems. The data followed a normal distribution, and the one-way analysis of variance (ANOVA) and post-hoc analysis by the Scheffé test were used to assess differences between histological groups. Only *p* values of < 0.01 were considered significant.

## 3. Results

The demographic and clinicopathological data of samples are reported in Table 1.

Immunohistochemical analysis revealed that TLR4 is strongly upregulated in non-tumoral epithelial surrounding OSCCs (Figure 1) compared to normal control epithelium, as well as in OSCC cells (Figure 2). TLR4 percentage of positive cells increases from normal oral control and non-lichenoid epithelium surrounding OSCC to lichenoid epithelium, reaching the highest value in OSCC EX-OLL (Figure 3A). Table 2 displays the TLR4 quantitative score (intensity of staining x percentage of positive cells) for all samples analyzed. 

In addition, the TLR4 staining location within the epithelium was assessed. TLR4 is located in the basal third of epithelium in non-lichenoid epithelium surrounding OSCC, while in lichenoid epithelium surrounding OSCC, TLR4 expression reached the intermediate layer (Figure 3B).

Statistically significant differences were assessed on quantitative scores among the pathological groups (Figure 4, ANOVA, *p* < 0.001). In detail, post-hoc analysis revealed that the quantitative score was statistically different between the LE-OSCC group and the NLE-OSCC group (Figure 4, *p* = 0.007), with the former being higher (63.11 ± 65.48), in comparison to NLES-OSCC (21.72 ± 54.80). Whereas the comparison of OSCC- ExNOLL group (155.17 ± 105.57) and the OSCC-ExOLL group (135.21 ± 87.32) did not detect statistical differences in their quantitative scores (Figure 4, *p* = 0.46). The correlation between the TLR4 quantitative score and the clinicopathological variables was performed (Table 3). The TLR4 score in the LES-OSCC samples from other anatomic sites was higher than in LES-OSCC from the tongue (*p* < 0.0001).

The downstream activation of the NFKβ pathway was detected by immunostaining of phospho-NFKβ in OSCC cells and in the surrounding non-tumoral epithelium of OSCC (Figure 5 and Figure 6). Cyclooxygenase-2 (Cox2) is also promoted by TLR4 signaling, and it was revealed that its expression was low in the surrounding non-tumoral epithelium of OSCC. However, it reached high levels in OSCC tumor cells. Interleukin 6 receptor staining, on the other hand, is abundant both in the non-tumoral epithelium surrounding OSCC and in tumor cells (Figure 5 and Figure 6).

## 4. Discussion

Inflammation has been shown to be pro-cancerogenic; however, the pathways that are responsible for this association are still under investigation. Toll-like receptors have been implicated as a key link between the tumor inflammatory environment and cancer promotion. To improve our knowledge on this subject, the expression of TLR4 in the epithelium tissue surrounding oral carcinomas was investigated. An increase in TLR4 immunoexpression was observed from normal oral mucosa to the epithelium surrounding OSCC to the tumor cells in OSCC. Moreover, the presence of a lymphocytic infiltrate in the surrounding mucosa of cancer samples was associated with increased TLR4 expression.

No statistical difference was observed when OSCC developed from oral lichenoid lesions versus non-oral lichenoid lesions. However, the results showed that the presence of a lichenoid infiltrate increases the number of TLR4-positive cells (Figure 3A), but not its intensity (Table 2), as well as increasing the number of positive epithelial layers, in the epithelium surrounding OSCC. Since an immunological response is a consequence of TLR4 activation, we hypothesized that OSCC developed from previous lesions with intense lichenoid cell infiltrate (lichenoid lesions, LL) would present higher TLR4 expression. The term OLL is used when it is not possible to fulfill both clinical and histopathologic criteria for oral lichen planus [20]. Previous reports have already shown that TLR4 is upregulated in OLL when compared to normal tissue [15,22]; however, its expression in OSCC developed from OLL had not yet been investigated. In OLP, Ge et al. (2012) demonstrated that the activation of TLR4 initiated a signaling cascade that resulted in the induction of NF-κB, which controls the release of proinflammatory cytokines and chemokines, suggesting that the TLR4 and NF-κB signaling pathway may be associated with the perpetuation processes of OLP [23]. Considering the importance of TLR4 in cancer pathogenesis, its expression in lichenoid lesions can result in malignant transformation susceptibility and increased tumor progression. It is important to highlight that TLR4 polymorphism is not associated with increased risk to OSCC [24,25]; therefore, its relation with cancer development must be caused by receptor activation and its downstream signaling cascade. Accordingly, we also detected the phosphorylation of NF-κB and the high expression of its downstream Cox-2 in tumor cells with TLR4 activation, suggesting that this signaling pathway still contributes to perpetuate proliferation signaling after cancer development, as observed by high levels of IL6R.

In OSCC, we observed high levels of positive TLR4 cells, especially in the presence of a lichenoid infiltrate. Szczepanski et al. (2009) showed that, in head and neck cancer, the presence of lipopolysaccharide (LPS) from bacteria resulted in NF-κB activation and inflammatory cytokine production that was dependent on TLR4 activation, resulting in tumor cell proliferation [26]. Considering that the oral cavity is home to a rich microbiota, and that oral cancer lesions are often colonized by bacteria and fungi, these pathogens, and their product LPS can trigger TLR4, which in turn releases cytokines, producing chronic inflammation and consequently promoting tumor growth. A relation between the anatomic site and the TLR4 score was detected in LES-OSCC samples, suggesting that anatomic sites with the propensity to accumulate bacteria biofilm may present higher TLR4 activation.

As in oral cancer, colorectal cancer (CRC) also develops in a microbe-rich environment. In CRC, the direct link between inflammation and cancer is well established and has been supported by epidemiological studies. TLRs have been implicated in CRC development and progression, and the contribution of TLR4 is considerably higher than the other TLRs [27]. TLR4 expression is required for dysplasia and polyp formation. In CRC, elevated TLR4 expression is observed in all tumor components, such as the epithelial, endothelial, and stromal layers [18], while TLR4 is expressed at a very low level in normal colorectal cells [28].

In the samples assessed in this study, we observed TLR4-positive cells in normal oral mucosa. TLR4-positive cells increased from normal samples to the surrounding epithelium and further increased in cancer samples, suggesting an important role of TLR4 in carcinogenesis. Similar to our findings, Mäkinen et al. (2016) [17], Szczepanski et al. (2009) [26], and Daskalopoulos et al. (2020) [29] also found TLR4 expression in normal mucosa. Moreover, the expression in normal tissues was lower than in the cancer counterpart.

Since TLR4 activation is associated with tumor progression [26], it is also important to analyze the relation of TLR4 with clinical-pathological parameters and prognosis markers. TLRs have been implicated on the invasive potential and aggressiveness of tumors [26,30,31,32]. According to Mäkinen et al. (2015), TLR4 expression was significantly associated with the invasiveness of, and also correlated with higher tumor grade of, early-stage oral tongue OSCC tumors [17]. Previously, Mäkinen et al. (2015) also showed that cytoplasmic TLR4 was stronger at the invasive front than on the surface of the tumor. The authors also showed that invading cells in an organotypic model expressed higher levels of TLR2 and TLR4 [33]. Kong et al. (2020) [32] revealed that TLR4 expression was related to lymphatic metastasis and to poor survival in OSCC. Although no significant correlation between TLR4 score and the specific groups assessed in this study was observed, when grouping all tumor samples, higher TLR4 levels were present on samples with lymph node metastasis. However, the high variability among the samples prevented statistical significance. Since we have observed that TLR4, and TLR4 expression on surrounding epithelium, may increase cell migration and invasion, further studies investigating the association of tumor recurrence and metastasis with TLR4 expression on surrounding epithelium are promising.

Immunotherapy has become a valuable treatment option against OSCC. Inhibition of the pathway between the programmed death 1 receptor (PD-1) and the programmed death ligand (PD-L1) decreases tumor growth, improving survival. However, such therapy appears to benefit only a subset of patients [34,35,36]. Therefore, the definition of OSCC originating from a lesion with phlogistic infiltrate, such as those originating from OLL, is very important in order to define subgroups, which will ensure a high number of TLR4-positive cells. TLR4 has been shown to exert immunosuppressive properties in different types of cancer [37,38]. Higher levels of TLR4 have already been correlated with high PD-L1 in non-small cell lung cancer [39], melanoma [40], and T-cell lymphomas [41]. The experimental inhibition of TLR4 was able to decrease PD-L1 expression on macrophages and dendritic cells co-cultured with OSCC cells [42]. Therefore, TLR4 level analysis may be a promising biomarker to define patients suitable for immune-targeted therapy.

## 5. Conclusions

Taken together, our results demonstrate TLR4 upregulation in OSCC, which can be associated with the cancer inflammatory microenvironment, since its expression increases when a lichenoid infiltrate surrounds the tumor. An understanding of the underlying mechanisms relating to TLR’s function in cancer is important because these are potential predictive markers that could assist in choosing treatment modalities. Due to its high levels of expression and its cell surface location, TLR4 is a potential target for a therapeutic approach in oral cancer.

## Figures and Tables

**Figure 1 biomolecules-12-00385-f001:**
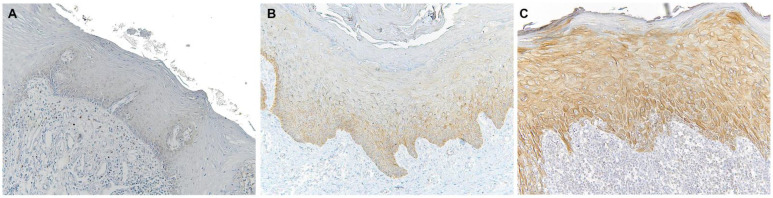
Microscopic images of TLR4 staining in representative lesions studied. (**A**), Normal oral mucosa. (**B**), Non-lichenoid epithelium surrounding OSCC (NLE-OSCC). (**C**), Lichenoid epithelium surrounding OSCC (LE-OSCC).

**Figure 2 biomolecules-12-00385-f002:**
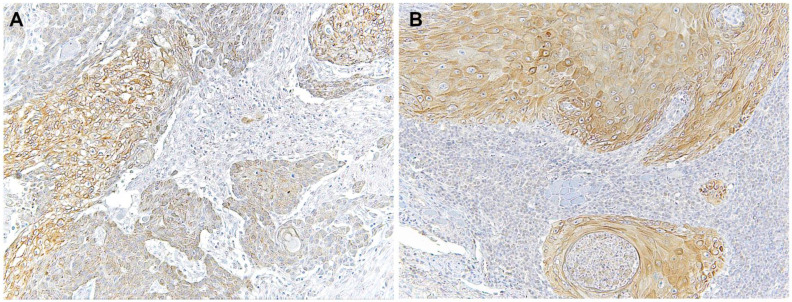
Microscopic images of TLR4 staining in representative lesions studied. (**A**), OSCC ex non-oral lichenoid lesion (OSCC Ex-NOLL). (**B**), OSCC ex oral lichenoid lesion (OSCC Ex-OLL).

**Figure 3 biomolecules-12-00385-f003:**
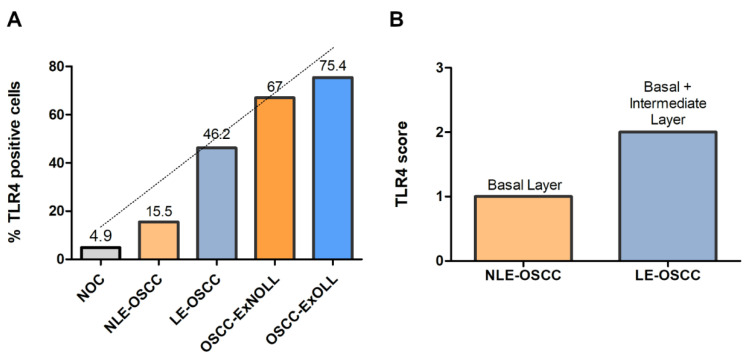
(**A**)**,** Mean percentage of TLR4-positive cells among the investigated lesions. (**B**), TLR4 score according to the epithelial layer in NLE-OSCC and LE-OSCC.

**Figure 4 biomolecules-12-00385-f004:**
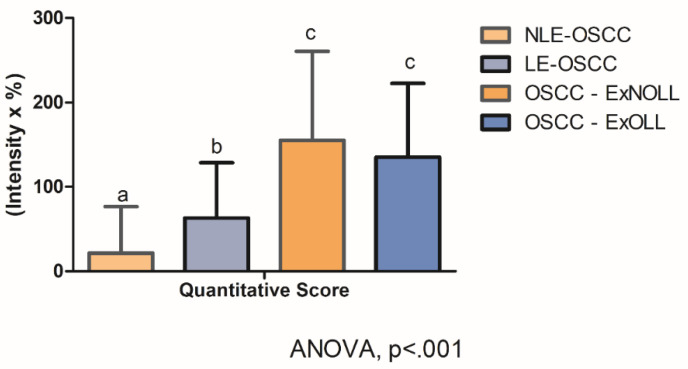
Statistical comparison among lesions regarding TLR4 staining quantitative score was statistically different (ANOVA, *p* < 0.001). Post-hoc analysis detected differences among the LE-OSCC group and the NLE-OSCC group (*p* = 0.007).

**Figure 5 biomolecules-12-00385-f005:**
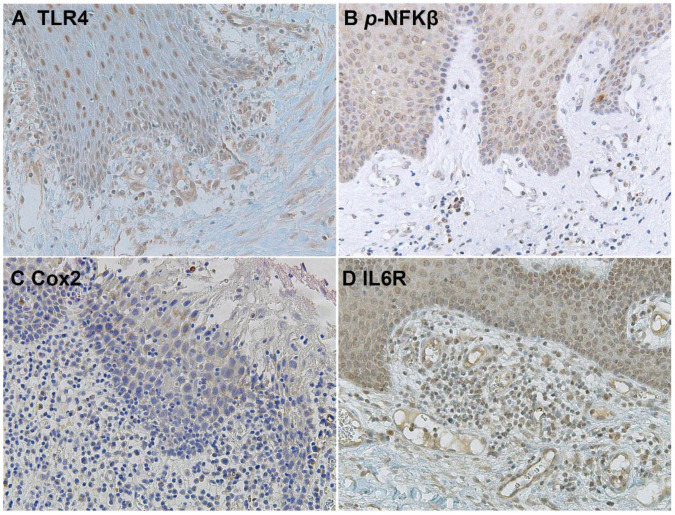
Microscopic images of inflammatory biomarkers in the surrounding epithelium of OSCC in representative lesions studied. (**A**), TLR4 staining. (**B**), *p*-NFKβ staining. (**C**), Cox2 staining. (**D**), IL6R staining.

**Figure 6 biomolecules-12-00385-f006:**
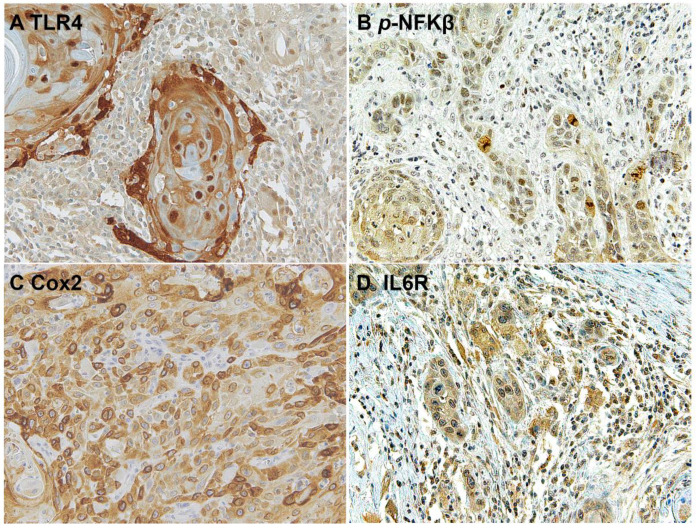
Microscopic images of inflammatory biomarkers in OSCC cells in representative lesions studied. (**A**), TLR4 staining. (**B**), *p*-NFKβ staining. (**C**), Cox2 staining. (**D**), IL6R staining.

**Table 1 biomolecules-12-00385-t001:** Clinicopathological data of samples.

		NOC (n = 30)	NLES-OSCC (n = 38)	LES-OSCC (n = 28)	OSCC-EX-OLL (n = 24)	OSCC-EX-NOLL (n = 30)
Age	Mean (±SD) years	59.4 ±15.3	64.1 ± 16.0	64.4 ± 14.6	64.7 ± 14.5	65.0 ± 16.8
Sex	Female	8 (26.6%)	9 (23.7%)	6 (21.4%)	7 (29.2%)	8 (26.6%)
	Male	22 (73.4%)	29 (76.3%)	22 (78.6%)	17 (70.8%)	22 (73.4%)
Anatomic site	Tongue	0	31 (81.6%)	24 (85.7%)	19 (79.2%)	25 (83.3%)
	Buccal mucosa	30 (100%)	3 (7.9%)	2 (7.1%)	2 (8.3%)	2 (6.6%)
	Trigonus	0	2 (5.2%)	0	1 (4.1%)	2 (6.6%)
	Hard palate	0	1 (2.6%)	1 (3.6%)	1 (4.1%)	1 (3.3%)
	Uvula	0	1 (2.6%)	0	0	0
	Gum	0	0	1 (3.6%)	1 (4.1%)	0
Grade	1	NA	9 (23.7%)	4 (14.3%)	9 (37.5%)	5 (16.6%)
	2	NA	12 (31.6%)	10 (35.7%)	10 (41.7%)	12 (40%)
	3	NA	6 (15.8%)	4 (14.3%)	2 (8.3%)	7 (23.4%)
	Unknown	NA	11 (28.9%)	10 (35.7%)	3 (12.5%)	6 (20%)
T stage	T1-T2	NA	21 (55.3%)	16 (57.1%)	18 (75%)	17 (56.7%)
	T3-T4	NA	2 (5.2%)	1 (3.6%)	1 (4.1%)	2 (6.6%)
	Unknown	NA	15 (39.5%)	11 (39.3%)	5 (20.9%)	11 (%)
N stage	Positive	NA	5 (13.2%)	6 (21.4%)	6 (25%)	5 (16.6%)
	Negative	NA	19 (50%)	13 (46.4%)	14 (58.3%)	15 (50%)
	Unknown	NA	14 (36.8%)	9 (32.1%)	4 (16.7%)	10 (33.4%)
M stage	Positive	NA	0	0	0	0
	Negative	NA	5 (13.2%)	2 (7.1%)	2 (8.3%)	4 (13.3%)
	Unknown	NA	33 (86.8%)	26 (92.9%)	22 (91.7%)	26 (86.7%)

SD, standard deviation.

**Table 2 biomolecules-12-00385-t002:** Quantitative score (intensity of staining × percentage of positive cells) of TLR4 staining in all samples investigated.

	Quantitative Score (Intensity x % Positive Cells)					
Sample	N	Mean	SD	Min	Max	Range
Normal Oral Mucosa Control (NOC)	30	3.53	3.84	0.0	15.0	15.00
Non-Lichenoid Epithelium Surrounding OSCC (NLES-OSCC)	38	21.72	54.80	0.0	300.0	300.00
Lichenoid Epithelium Surrounding OSCC (LES-OSCC)	28	63.11	65.48	0.0	255.0	255.00
OSCC Ex Non-Lichenoid Lesion	30	155.17	105.57	0.0	300.0	300.00
OSCC Ex Lichenoid Lesion	24	135.21	87.32	0.0	300.0	300.00
Total	169	68.85	90.18	0.0	300.0	300.00

SD, standard deviation.

**Table 3 biomolecules-12-00385-t003:** Correlation of TLR4 quantitative score (mean ± SD) with clinicopathological variables.

		NLES-OSCC	LES-OSCC	OSCC-EX-OLL	OSCC-EX-NOLL	All Samples
Grade of differentiation	G1	60.17 (±103.6)	3.87 (±5.92)	178.6 (±112.3)	156.3 (±144.1)	101.3 (±120)
	G2	11.91 (±19.13)	64.45 (±40.93)	124 (±64.37)	181.7 (±115.6)	97.57 (±95.93)
	G3	6.33 (±13.22)	58.25 (±56.07)	200 (±70.71)	122.5 (±63.54)	78.11 (±80.26)
	*p*	0.17	0.064	0.33	0.56	0.73
T status	T1 + T2	30.73 (±73.02)	34.86 (±37.16)	139.3 (±92.08)	177.5 (±106.2)	92.81 (±103)
	T3 + T4	5 (±7.07)	132 (±0)	200 (±0)	112.5 (±159.1)	94.5 (±104.5)
	*p*	0.63	NA	NA	0.44	0.96
N status	N0	31.5 (±71.82)	28.8 (±41.68)	164.2 (±91.27)	143.2 (±103)	86.39 (±100.2)
	N+	9.8 (±8.63)	66.33 (±35.79)	128.3 (±87.79)	220.8 (±119.4)	110.5 (±107.4)
	*p*	0.51	0.088	0.43	0.1574	0.34
Anatomical site	Tongue	22.24 (±59.64)	43.75 (±42.87)	152.1 (±86.7)	160.2 (±103.8)	87.22 (±97.54)
	Others	19.43 (±26.93)	179.3 (±59.31)	71 (±59.2)	130 (±123.3)	88.48 (±91.16)
	*p*	0.90	<0.0001	0.063	0.56	0.95

## Data Availability

Not applicable.

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
