# Peer review of "TLR4 Expression in Ex-Lichenoid Lesions—Oral Squamous Cell Carcinomas and Its Surrounding Epithelium: The Role of Tumor Inflammatory Microenvironment"

_biomolecules, 2022, doi:10.3390/biom12030385_

Round 1
Reviewer 1 Report
With the work by Visioli et al. “TLR4 Expression in Ex-Lichenoid Lesions - Oral Squamous Cell Carcinomas and Its Surrounding Epithelium: the Role of Tumor Inflammatory Microenvironment” the authors tried to analyze the inflammatory status within the tumor microenvironment and also in the tumor surrounding tissue based on TLR expression status. This is an important issue, since inflammation can drive cancerogenic processes in many tumor entries and TLR ligation induces HNSCC-derived tumor cell proliferation.
The manuscript is written well. However, some points need to be noted:
- As the authors stated, TLR4 expression was already studied on oral cancer as well as the consequences of TLR4 activation for oral cancer cell proliferation. Also, TLR4 expression on oral lichenoid lesions was described. Therefore, some results of the study are confirmative. A new aspect of the present study is the analyses of TLR4 expression in OSCC developing from lichenoid lesions versus non-lichenoid lesions. Since the differences in TLR4 expression between OSCC-ExNOLL and OSCC-ExOLL are not statistically significant, the authors' interpretation of the data is not supported by their results.
- The authors describe an association of TLR4 ligation and NF-kB activation. Why the authors did not analyze e.g. nuclear phospho-p65 in their samples?
- Correlation analyzes with clinical parameters are missing
Minor points:
- The manuscript has some typing mistakes
- some recent reports concerning TLR4 and oral cancer are missing
Author Response
Reviewer #1
With the work by Visioli et al. “TLR4 Expression in Ex-Lichenoid Lesions - Oral Squamous Cell Carcinomas and Its Surrounding Epithelium: the Role of Tumor Inflammatory Microenvironment” the authors tried to analyze the inflammatory status within the tumor microenvironment and also in the tumor surrounding tissue based on TLR expression status. This is an important issue, since inflammation can drive cancerogenic processes in many tumor entries and TLR ligation induces HNSCC-derived tumor cell proliferation.
The manuscript is written well. However, some points need to be noted:
- As the authors stated, TLR4 expression was already studied on oral cancer as well as the consequences of TLR4 activation for oral cancer cell proliferation. Also, TLR4 expression on oral lichenoid lesions was described. Therefore, some results of the study are confirmative. A new aspect of the present study is the analyses of TLR4 expression in OSCC developing from lichenoid lesions versus non-lichenoid lesions. Since the differences in TLR4 expression between OSCC-ExNOLL and OSCC-ExOLL are not statistically significant, the authors' interpretation of the data is not supported by their results.
Response: We are thankful to the reviewer for raising this issue. We have improved the discussion to address this data and we have modified the text regarding the interpretation of data.
- The authors describe an association of TLR4 ligation and NF-kB activation. Why the authors did not analyze e.g. nuclear phospho-p65 in their samples?
Response: We have performed phospho-NF-KB immunostaining in selected cases, and as expected, high TLR4 levels were followed by activation of NF-KB indicated by high levels of positive cells, indicating that this inflammatory pathway is activated in these lesions. Please, see new figures 5 and 6.
- Correlation analyzes with clinical parameters are missing
Response: We are thankful to the reviewer for this important recommendation. The correlation of TLR4 score with clinicopathological data was performed and is available at new table 3.
Minor points:
- The manuscript has some typing mistakes
Response: We are thankful to the reviewer for pointing out this issue. The manuscript has been reviewed and the typos were corrected.
- some recent reports concerning TLR4 and oral cancer are missing
Response: As recommended by the reviewer, recent studies about TLR4 in oral cancer were added to the manuscript. The following references were added to the discussion section:
de Barros, G.C.; et al. Toll-like receptor 2 rs4696480 polymorphism and risk of oral cancer and oral potentially malignant disorder. Arch Oral Biol. 2017 Oct;82:109-114. doi: 10.1016/j.archoralbio.2017.06.003. Epub 2017 Jun 6. PMID: 28624699.
Zeljic, K.; et al. Association of TLR2, TLR3, TLR4 and CD14 genes polymorphisms with oral cancer risk and survival. Oral Dis 2014, Volume 20, p. 416-24. doi: 10.1111/odi.12144.
Daskalopoulos, A.G.; et al. Assessment of TLR4 and TLR9 signaling and correlation with human papillomavirus status and histopathologic parameters in oral tongue squamous cell carcinoma. Oral Surg Oral Med Oral Pathol Oral Radiol 2020, Volume 129, p. 493-513. doi: 10.1016/j.oooo.2020.01.001.
Kong, Q.; et al. Autophagy inhibits TLR4-mediated invasiveness of oral cancer cells via the NF-κB pathway. Oral Dis 2020. doi: 10.1111/odi.13355.
Reviewer 2 Report
Dear Authors,
You have nicely presented your paper. You could add a bit more in relevance to TLR4 ligands in your introduction and the alternative activation pathways (MyD88 and non-MyD88) along with its downstream effectors.
Author Response
Reviewer #2
You have nicely presented your paper. You could add a bit more in relevance to TLR4 ligands in your introduction and the alternative activation pathways (MyD88 and non-MyD88) along with its downstream effectors.
Response: We are grateful to the reviewer for this important suggestion. We have modified the introduction section to include more information about TLR4 ligands, the activation pathways, and downstream effectors.
Please, see revised introduction (Pages 2 and 3):
“The modulation of the inflammatory process by Toll-Like Receptors can be a key factor linking inflammation and tumor development [10]. Toll-like receptors (TLRs) are expressed on cells of the immune system as dendritic cells, as well as on non-immune cells such as keratinocytes. TLRs act as sensors to recognize pathogens and can regulate innate and adaptive immune responses, recruiting immune cells [11,12].
A total of 10 TLRs have been identified in humans, from these TLR4 has been associated with pro-tumorigenic outcomes [13, 14]. Toll-like receptor 4 (TLR4) primarily recognizes and is activated exogenously by lipopolysaccharide (LPS). However, endogenous ligands from damaged tissue as high-mobility group box 1 (HMGB1), heat shock proteins (HSP), reactive oxygen species (ROS), peptides derived from fibrinogen, and monosodium urate crystals were also identified [10,11].
TLR4 activation requires interaction with the auxiliary protein CD14 and the co-receptors myeloid differentiation protein 2 (MD-2), and downstream signaling cascade is mediate through two different pathways: myeloid differentiation factor 88 (MyD88)-dependent and MyD88-independent. The MyD88 axis results on activation of the NF-Kβ signaling and, consequently, of a inflammatory pathway with COX-2 activation and secretion of different cytokines and growth factors, such as IL‐1β, IL‐6, IL‐8, and TNF‐ α, that will induce a pro-angiogenic, pro-survival and pro-proliferation response. The MyD88-independent axis, on the other hand, recruits the TRAM adapter which activates TRIF. TRIF with TBK1 and IKKi proteins phosphorylates IRF3 and activates this transcription factor resulting in the transcription of Type I Interferons. [10-14].”
Reviewer 3 Report
This study demonstrated very interesting data but I think that it is not possible to interprete data on base of the detection of only 1 protein! It is known some cooperation of TLR4 with CD14. It should be very interesting to show expression of this protein. Authors interprete their finding by the inflammatory microenvironment in cancer. I strongly support detection of factors such as IL-6, IL-8 and TGF-β (some member of fasmily) to demonstrate this inflammation supporting micromilieu. I strongly reccommend to resubmit article after this revision.
Author Response
Reviewer #3
This study demonstrated very interesting data but I think that it is not possible to interprete data on base of the detection of only 1 protein! It is known some cooperation of TLR4 with CD14. It should be very interesting to show expression of this protein. Authors interprete their finding by the inflammatory microenvironment in cancer. I strongly support detection of factors such as IL-6, IL-8 and TGF-β (some member of fasmily) to demonstrate this inflammation supporting micromilieu. I strongly reccommend to resubmit article after this revision.
Response: We are grateful to the reviewer for this important suggestion. We have performed phospho-NFKB, COX-2 and IL6R staining in selected cases, and as expected high TLR4 levels were followed by activation of NFKB as indicated by high levels of positive cells, as well as its downstream COX-2 and IL6R indicating that this inflammatory pathway is activated in these lesions. Please, see new figures 5 and 6.
Round 2
Reviewer 1 Report
All of my concerns have been addressed and satisfactorily answeredReviewer 3 Report
Revised manuscript is suitable for acceptance